# Novel Technique to Measure Pulse Wave Velocity in Brain Vessels Using a Fast Simultaneous Multi-Slice Excitation Magnetic Resonance Sequence

**DOI:** 10.3390/s21196352

**Published:** 2021-09-23

**Authors:** Ju-Yeon Jung, Yeong-Bae Lee, Chang-Ki Kang

**Affiliations:** 1Department of Health Science, Gachon University Graduate School, Gachon University, Incheon 21936, Korea; 9955me@gachon.ac.kr; 2Neuroscience Research Institute, Gachon University, Incheon 21565, Korea; 3Department of Neurology, Gil Medical Center, Gachon University College of Medicine, Incheon 21565, Korea; 4Department of Radiological Science, College of Health Science, Gachon University, Incheon 21936, Korea

**Keywords:** pulse wave velocity, magnetic resonance imaging, stroke, simultaneous multi-slice, keyhole technique

## Abstract

In this study, we proposed a novel pulse wave velocity (PWV) technique to determine cerebrovascular stiffness using a 3-tesla magnetic resonance imaging (MRI) to overcome the various shortcomings of existing PWV techniques for cerebral-artery PWV, such as long scan times and complicated procedures. The technique was developed by combining a simultaneous multi-slice (SMS) excitation pulse sequence with keyhole acquisition and reconstruction (SMS-K). The SMS-K technique for cerebral-artery PWV was evaluated using phantom and human experiments. In the results, common and internal carotid arteries (CCA and ICA) were acquired simultaneously in an image with a high temporal resolution-of 48 ms for one measurement. Vascular signals at 500 time points acquired within 30 s could generate pulse waveforms of CCA and ICA with 26 heartbeats, allowing for the detection of PWV changes over time. The results demonstrated that the SMS-K technique could provide more PWV information with a simple procedure within a short period of time. The procedural convenience and advantages of PWV measurements will make it more appropriate for clinical applications.

## 1. Introduction

A stroke is a neurological deficit caused by vascular injury in the central nervous system and the second leading cause of mortality worldwide. It is also responsible for a variety of diseases, including permanently acquired disability, late-onset dementia, and epilepsy in elderly individuals [1]. The burden of a stroke is constantly increasing in terms of mortality, morbidity, and disability; thus, various studies for overcoming a stroke are actively attempted all over the world. Recent studies have shown that methods for risk factor detection and stroke prevention are of great importance [2,3,4,5,6].

Cardiovascular diseases such as hypertension, heart failure, and atrial fibrillation, which are known as the major risk factors for stroke [7], are affected by vascular stiffness. Pulse wave velocity (PWV), which can quantitatively measure the elasticity of the arteries for vascular stiffness, is considered an important clinical indicator for vascular aging and the early diagnosis of vascular diseases [8,9]. The loss of arterial elasticity can reduce the ability to store and supply blood to distal tissues and is closely related to cardiovascular diseases, including hypertension [10], coronary artery disease [11] and heart failure [12].

PWV is closely associated with a stroke [11] and is also used to evaluate treatment effectiveness and risk stratification [13,14]. Although the importance of PWV is increasing, the existing method of measuring PWV is not suitable for cerebral-artery PWV; thus, PWV studies for predicting a stroke are still lacking.

The most common clinical approach for PWV measurement is tonometry, which measures pressure waveforms at two locations (the carotid and femoral arteries) to provide a global PWV value. Although this technique can estimate the whole vascular PWV, it has limitations in estimating local vascular PWV or cerebral-artery PWV. There is also a Doppler that measures the PWV through the changed frequency in the blood flow by applying ultrasound to a blood vessel, but this has limitations in estimating cerebral-artery PWV due to the skull. Although it is possible to measure cerebral-artery PWV using the transcranial Doppler (TCD) technique [15], the application of cerebral-artery PWV measurement remains limited due to low ultrasound transmittance and operator experience, which may be a limiting factor for widespread diagnosis using TCD.

Magnetic resonance imaging (MRI)-based PWV measurements are developed to overcome the limitations of tonometer and Doppler techniques [16]. MRI-based PWV measurements using phase-contrast (PC) MRI sequences can allow for quantitative hemodynamic evaluation, including vascular structure and flow rate information [17]. The techniques were developed from 2-dimensional (2D) to 4D PC MRI in order to provide highly accurate measurements of cerebral-artery PWV with flow velocity and an accurate distance from 3D vascular anatomy [18]. However, the limitation of this flow imaging is its long scan time [18,19], and since the two target vessels are acquired as separate scans, the ensemble of heartbeats is different, which requires an additional process to synchronize the blood transit times. Furthermore, only one waveform for vessel pulsation was used to calculate the PWV for vascular stiffness.

Therefore, a novel technique of PWV, compensating for various limitations, is needed for widespread clinical use, especially for the prevention and diagnosis of cerebral vascular diseases. This study is a preliminary study to develop a new PWV measurement technique that can measure the stiffness of cerebral blood vessels and prove its feasibility. In this study, we developed a new technique for PWV using a simultaneous multi-slice technique (SMS), which takes two target images simultaneously, without the need to synchronize blood transit times, and a keyhole technique that collects dynamic blood flow signals while significantly reducing scan times. In addition, the characteristics of the technique proposed for PWV measurements (SMS-K), combining the simultaneous multi-slice excitation technique with the keyhole technique, were evaluated to determine its clinical usefulness.

## 2. Materials and Methods

### 2.1. Method of SMS-K

SMS-K is an application of the K-space data acquisition method. In MRI, K-space is the Fourier-transformed (FT) spatial frequency of a two-dimensional (2D) image (space). MRI images were firstly acquired in the form of spatial frequencies and then reconstructed through their inverse FT. Therefore, K-space data are referred to as the raw data of MRI images.

For the cerebral-artery PWV measurement, an SMS technique, which was previously introduced for fast-imaging [20,21], was combined with a keyhole technique that consisted of only a few K-space center lines [22,23]. At first, for the simultaneous acquisition of an image for two different locations, the SINC-shaped radio-frequency pulse was modulated with a cosine function which was the same as the conventional SMS method [20]. Using the developed SMS method, an image of two target blood vessels was acquired and evaluated through phantom and in vivo human images.

For the keyhole acquisition and reconstruction of an image with resolution 256 × 256, a K-space dataset of 256 lines along the phase-encoding direction (one of spatial frequency axes) is necessary, which can be referred to as the reference K-space dataset in the keyhole technique; multiple datasets with a few K-space center lines were acquired for fast imaging [22,23]. In this study, a 3-line K-space dataset was acquired and shared the other peripheral K-space lines with the reference K-space dataset in reconstruction (Figure 1). Then, the SMS-K was evaluated with in vivo human image.

All experiments were conducted using a 3-tesla (3 T) MRI scanner (Siemens Verio, Erlangen, Germany) equipped with a 12-channel radio-frequency head matrix coil.

### 2.2. Phantom MRI Experimental Protocols

A phantom experiment was performed to test the SMS-K for PWV measurements. The phantom was created using a model of the internal carotid artery (ICA) and common carotid artery (CCA) with an inner diameter of 6 mm and a total length of 20 cm (Figure 2A). The process of 3D printing was performed using 3DWOX 2X (Sindoh, Seoul, Korea) based on fused deposition modeling, and the phantom was modeled using 3DWOX Desktop slicer software (Sindoh, Seoul, Korea).

A polylactic acid filament (Sindoh, Seoul, Korea) with a diameter of 1.75 mm and a nozzle diameter of 0.4 mm were used for the phantom modeling. A 100% linear filling pattern was used to prevent the leakage of fluid and control the pressure. For MRI imaging, diluted gadolinium was included in the phantom model.

Phantom imaging was performed using 3D time-of-flight (TOF) MR angiography and 2D gradient echo (GRE) sequences to localize and capture anatomical reference images with 256 × 256 resolution, including ICA and CCA slices, respectively (11 slices with 3 mm thickness and 10 mm gap between slices). The SMS-K sequence with full K-space lines was applied to acquire the target ICA and CCA slices simultaneously into one slice image, and the image was compared with GRE images (Figure 2B,C). Furthermore, the SMS-K image with full K-space lines was used as a reference image to reconstruct the keyhole datasets with 3 K-space lines (Figure 1A,B). Table 1 presents the MR parameters for the phantom experiment.

### 2.3. Human MRI Experimental Protocols

A healthy subject participated in this study for the cerebral-artery PWV measurements after obtaining written informed consent prior to the experiment. A 3D sagittal TOF magnetic resonance angiography (MRA) was performed for the anatomical structures of the target blood vessels (CCA and ICA). GRE and SMS-K sequences were used to acquire the target vessel images, and then directly compared to each other (Figure 3).

After determining the exact target locations, radio-frequency pulses from SMS-K simultaneously excited two slices where the selected vessel segments, separated with a distance of 60 mm, were present. The SMS-K dataset including the entire K-space lines and with a scan time of approximately 4 s, was used as reference data for keyhole reconstruction. Note that an anatomic reference image for the vessels at two different locations can be acquired when the option for keyhole acquisition is deselected and the number of measurements is set to 1.

Blood signal waveforms in CCA and ICA segments were acquired after reconstructing SMS-K datasets containing three K-space lines, which took 48 ms because a repetition time (TR) of 16 ms was used. Furthermore, 500 measurements were taken to estimate the blood flow changes over time. Table 1 presents the MR parameter for the human experiment.

### 2.4. Human Heart-Finger PWV Experiment Protocols

In comparison with the SMS-K data, the global PWV was measured using a heart-finger PWV (STD-1000, Anyang, Korea). The STD-1000 was composed of four electrocardiogram (ECG) sensors and an SpO2 sensor. ECG data were acquired from the pulse wave of the bilateral radial arteries in the wrists and posterior tibial arteries of the ankles. A pulse wave at the left index finger attached to the SpO2 sensor was acquired using photoelectric plethysmography (PPG). To calculate PWV, the pulse transit time was obtained from the delay between an R-wave of ECG and a peak signal of PPG, as well as the spatial distance of the heart to the index finger, inferred by using the subject’s height (height × 0.56). The obtained mean pulse transit time and distance were 226 ms and 99.7 cm, respectively. The final PWV was calculated using the following formula:$$\mathrm{Pulse}\;\mathrm{wave}\;\mathrm{velocity}\;\left(\mathrm{PWV}\right)=\;\frac{\mathrm{Distance}\;\left(\Delta \;d\right)}{\;\mathrm{Pulse}\;\mathrm{Transit}\;\mathrm{Time}\;\left(\Delta \;t\right)}$$

### 2.5. Data Analysis

For a vascular signal analysis using multiple PWVs, the masks of the CCA and ICA vessels were generated from the SMS reference image using the MRIcron program [24]. To extract the signal waveforms of CCA and ICA, the masks were applied to 500 SMS-K images which were reconstructed by sharing with the peripheral lines of the reference K-space dataset. The 500 signals extracted every 48 ms constituted the pulse waveforms of the ICA and CCA. The acquired signals were interpolated by a cubic spline with 12 points in order to obtain more accurate peak signals in the pulse waveforms. All the image and signal processing were performed in MATLAB software (R2018b version, Mathworks, Natick, MA, USA) [25].

## 3. Results

The CCA and ICA vascular phantom was used to determine whether the developed SMS-K sequence could take exactly two target sections in one slice image (Figure 2). We found that CCA and ICA vessels were positioned at the same location in both the SMS-K image (Figure 2C) and the mean image (Figure 2B) of the 3rd slice (CCA target segment) and the 9th (ICA target segment) slice of 2D GRE images. During a human experiment, we also found the same result when comparing the SMS-K image and the mean image of 2D GRE slices (Figure 3).

In the SMS-K experiment for PWV, a total scan time of 24 s was required for 500 keyhole acquisition images, resulting in 26 cardiac pulses from each vessel. This corresponds to an average heart rate of approximately 920 ms/cycle. The 26 peak signals through the pulse waveforms were used for further analysis measuring the average pulse transit time between two vessels, CCA and ICA, wherein each transit time was calculated using the time differences between neighboring peaks of two vessels. In this result, the mean transit time and the calculated PWV between CCA and ICA were 25.23 ± 6.57 ms and 2.38 m/s for 26 peaks, respectively (Figure 4A).

In addition, we analyzed sectional pulse waveforms to examine changes in PWV over time. Each section consisted of 100 images (points) for 4.8 s (Figure 4B). The average transit times were calculated to measure the mean PWV (Figure 4C). The mean peak transit times and PWVs of the sections were 32.0 ms, 29.6 ms, 28.0 ms, 16.8 ms and 19.2 ms, and 1.88 m/s, 2.03 m/s, 2.14 m/s, 3.57 m/s and 3.13 m/s, respectively (Table 2). The maximum PWV (3.57 m/s) and minimum transit time (16.8 ms) appeared in the 4th section. On the other hand, the minimum PWV (1.88 m/s) and maximum transit time (32.0 ms) appeared in the 1st section. Note that the heart-finger PWV obtained for the comparison with the cerebral PWVs was 4.41 m/s.

## 4. Discussion

This study aimed to measure cerebral-artery PWV using a new fast-imaging method combined with SMS and keyhole techniques (SMS-K). The proposed SMS-K technique for cerebral-artery PWV was evaluated using phantom and human experiments.

By analyzing the SMS-K data, it can be seen that this technique has various advantages over the existing PC MRI techniques. First, PC-based PWV techniques require higher sampling points for accurate PWV measurements. However, high sampling points are accompanied by a low temporal resolution and long scan time. Therefore, it may be one of the major limitations of clinical applications when performing PC-based PWV measurements [26].

Second, PC-based techniques require additional PPG data for cardiac gating that is synchronized with phase-encoding gradients. Thus, only one pulse wave is acquired at a scan time of 10 min or more [18,19]. In addition, compensation for motion caused by longer scan times should also be considered [26].

Third, it is difficult to perform real-time PWV fluctuation analysis using the PC-based PWV measurement technique because it can only generate a single pulse wave for PWV. Finally, because some PC-based techniques use two different vascular pulse waves acquired at different times in the two target vessels, additional data processing is required to calculate the transit time between the peaks of the two vascular pulses.

The present SMS-K technique has advantages in terms of temporal resolution and scan time. Furthermore, since two different vascular images are acquired at the same time in this SMS-K technique for PWV measurement, a PPG acquisition and a procedure for cardiac gating are not necessary. The additional main feature of SMS-K, which enables fast scanning using the keyhole technique, is that it can analyze real-time changes in PWVs, which can measure dynamic vascular elasticity over time.

In this study, we obtained 500 temporal data points in a fast scan time using the SMS-K technique. In contrast to the existing PC-based MRI techniques for PWV, a multiple pulse waveform, which is 26 pulses over 24 s, and 26 peaks to calculate the pulse transit time between the two vessels were acquired in this SMS-K and could be utilized for further analysis. Based on these characteristics, it is expected that a novel diagnosis of the elasticity of blood vessels, that is, vascular stiffness, is possible.

The average cerebral-artery PWV of the sections is 2.55 ± 0.75 m/s, which is almost the same as that of 2.86 m/s in the previous PC-based CCA-ICA PWV study [27]. Furthermore, this study can provide different PWV values depending on specific situations or conditions, including adaptation to loud scanning noise within the MRI environment [28]. The SMS-K technique is expected to lead to a more convenient and quick diagnosis of cerebrovascular stiffness, as it can provide dynamic vascular changes according to the heartbeat.

According to a previous study, PWV values may vary with equipment such as tonometry and Doppler [29]. Although the correlation of the values between devices is high, the PWV magnitudes may vary from device to device. Therefore, the standard PWV value must be further evaluated according to the equipment and system.

In this study, heart-finger PWV in a healthy subject was similar to the previous global PWV [30]. However, since this value was different from that of cerebral-artery PWV between CCA and ICA, appropriate attention is necessary when assessing vascular stiffness in cerebral-artery PWV using devices that measure global PWV. The SMS-K technique for cerebral-artery PWV should be accompanied by further studies to establish it as a major tool for judging the health status of cerebral blood vessels. With the aid of further studies, the SMS-K technique could become an optimal method for local cerebral-artery PWV.

Although SMS-K has various advantages in this study, some limitations need to be addressed. During PWV analysis in the left vessels, the CCA and ICA were located along the vertical direction; thus, two vascular signals were projected onto the same location in the SMS image, resulting in indistinguishable signals between the two vessels. This limitation also affected the right CCA signal; thus, the unwanted signal might cause the pulse waveform to not be as sharp as shown in Figure 4. Therefore, an additional function to separate the two vascular signals must be added, which can be resolved using image rotation during acquisition (Appendix A). In addition, the distance between the two slices used in PWV calculation was not accurate because the distance (Δd) between the two target vessels was not calculated from the actual vascular structure with curvature. Therefore, vessel segmentation techniques should be used to calculate the actual vascular length between target vessels. Finally, the proposed SMS-K method needs further evaluated with more subjects, and further quantitative evaluation and technical improvement should be followed for clinical applications.

## 5. Conclusions

In this study, we developed a new technique that could obtain PWVs over time by using multiple pulse waveforms, and conveniently measure cerebral-artery PWV while compensating for the shortcomings of existing PWV methods. The technique included the following: the SMS technique enabled the simultaneous acquisition of information from two separate locations, and the keyhole technique enabled fast image acquisition and reconstruction. This allowed the evaluation of the real-time changes in cerebral-artery PWV, as multiple PWV signals could be calculated, in spite of a simplified procedure and greatly reduced scan time. This, in turn, enabled increased clinical applications. Proving the effectiveness of the technique through many additional studies in the future will provide a good opportunity for research in the field of cerebral-artery PWV, which has not been studied much until now, and is expected to make a great contribution to the prevention of cerebrovascular diseases, such as the stroke. It is expected that it could be used as a representative method for measuring cerebral-artery PWV and as a new biomarker for vascular disease research.

## Figures and Tables

**Figure 1 sensors-21-06352-f001:**
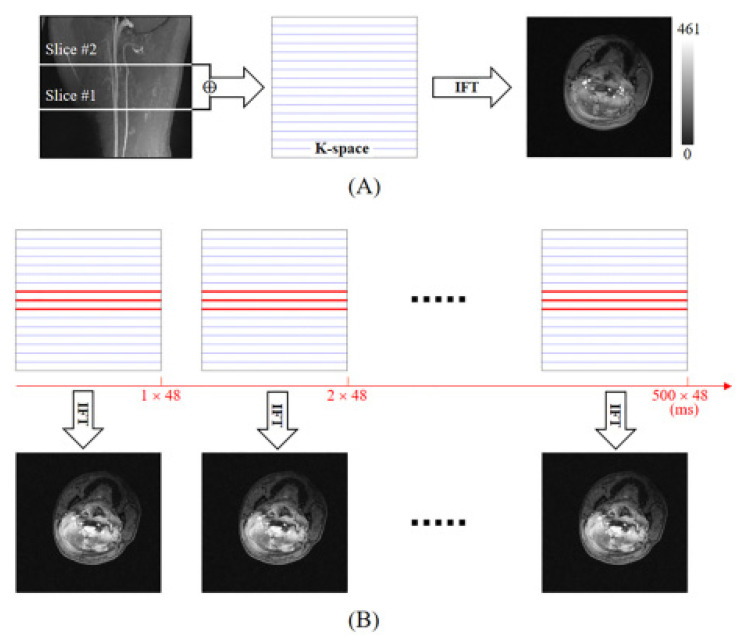
Conceptual diagram of SMS-K technique. (**A**) The reference SMS-K image with full K-space acquisition lines containing all information from both slices (3rd and 9th). (**B**) The 500 SMS-K measurements using keyhole acquisition and reconstruction technique. Because one measurement acquires three K-space center lines (red thick lines), a scan time of 48 ms is required. Note that repetition time (TR) is 16 ms. Other K-space information from the reference SMS-K data was used for the keyhole reconstruction (blue thin lines). A gray scale bar indicates the signal intensity. Abbreviations: SMS-K, simultaneous multi-slice keyhole technique; IFT, inverse Fourier transform.

**Figure 2 sensors-21-06352-f002:**
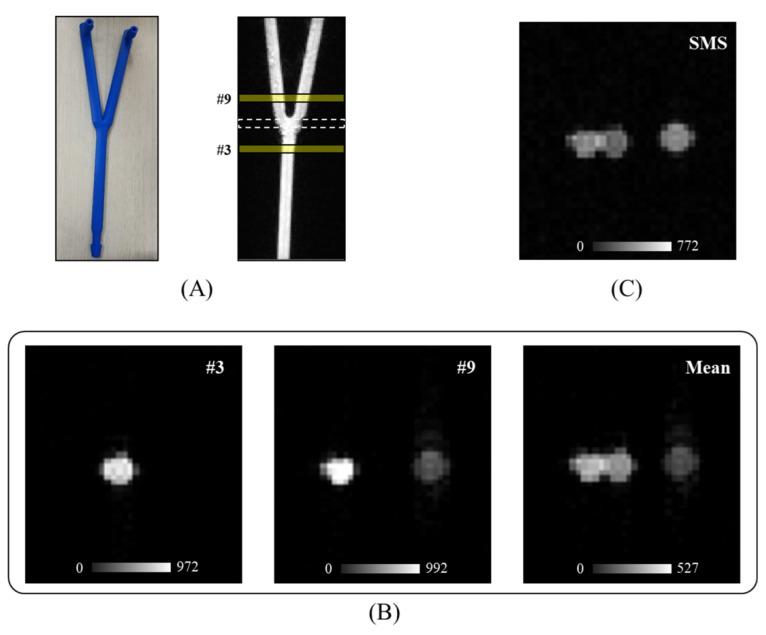
Phantom images comparing the SMS-K image with the mean GRE image of two selected slices. (**A**) The 3D modeling image and the phantom image acquired with 3D TOF MRA. (**B**) GRE images and the mean image of the 3rd and 9th slices. (**C**) SMS-K image acquired simultaneously on the 3rd and 9th slices. Gray scale bars indicate the signal intensity. Abbreviations: 3D TOF MRA, 3-dimensional time-of-flight magnetic resonance angiography; GRE, gradient echo; SMS-K, simultaneous multi-slice keyhole technique; 3D, 3 dimensional.

**Figure 3 sensors-21-06352-f003:**
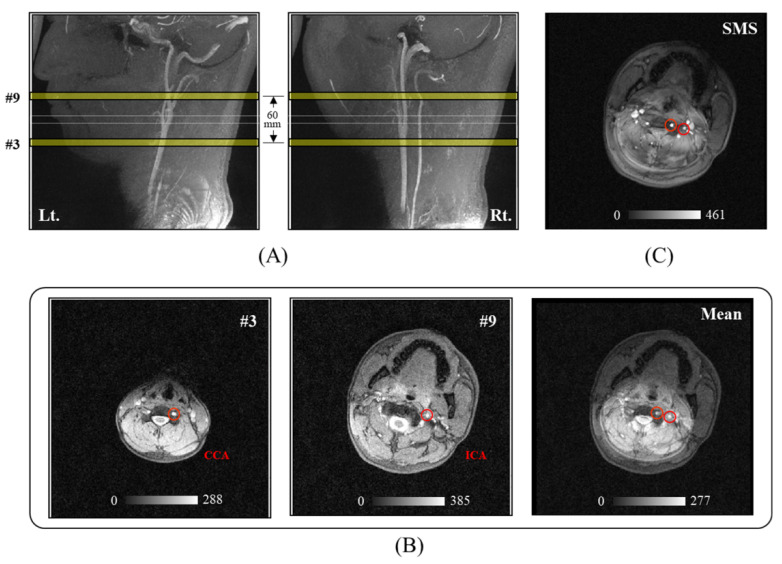
Human images comparing the SMS-K image with the mean GRE image of two selected slices. (**A**) Maximum-intensity projection images for CCA and ICA acquired with 3D TOF MRA. (**B**) GRE images and the mean image of the 3rd and 9th slices. (**C**) SMS-K image acquired simultaneously on the 3rd and 9th slices. Red circles indicate the target vessels for further analysis. Gray scale bars indicate the signal intensity. Abbreviations: 3D TOF MRA, 3-dimensional time-of-flight magnetic resonance angiography; CCA, common carotid artery; GRE, gradient echo; ICA, internal carotid artery; Lt, left; Rt, right; SMS-K, simultaneous multi-slice keyhole technique.

**Figure 4 sensors-21-06352-f004:**
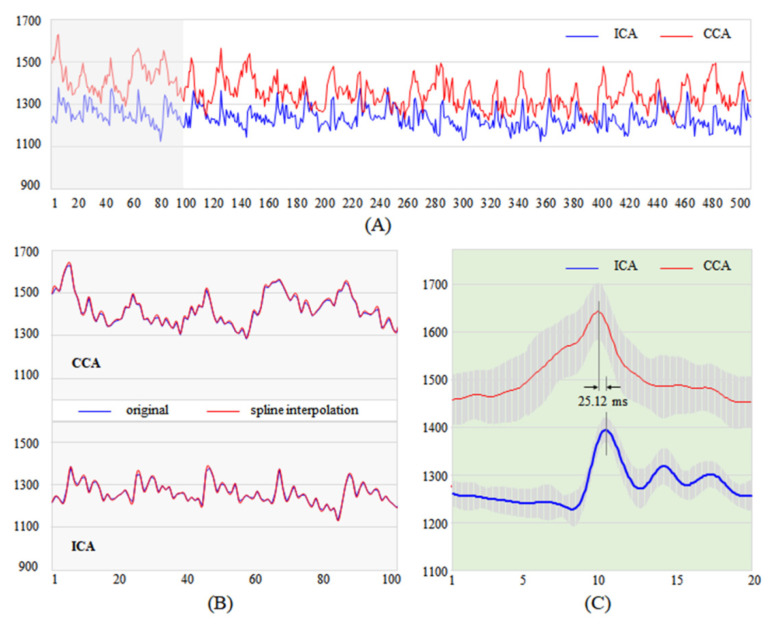
Pulse waveforms of CCA and ICA. (**A**) CCA (red) and ICA (blue) pulse waveforms consisted of 500 measurements for 24 s using SMS-K technique. (**B**) The pulse waveforms (blue) for the section of 100 measurement and the interpolated waveforms (red) of CCA and ICA. (**C**) Average pulse waveforms. The mean transit time of 5 sections between CCA and ICA was presented. Abbreviations: CCA, common carotid artery; ICA, internal carotid artery; SMS-K, simultaneous multi-slice keyhole technique.

**Table 1 sensors-21-06352-t001:** MR parameters for cerebral-artery PWV.

	Phantom	Human
MRI (Coil Channel)	3 T (12 ch.)	3 T (12 ch.)
Scanning Sequence	2D GRE	2D GRE
TR (ms)	20	16
TE (ms)	4	4.14
FA (°)	9	18
Resolution (mm^3^)	1.0 × 1.0 × 5.0	0.5 × 0.5 × 3.0
Matrix size	256 × 256	256 × 256
Bandwidth (Hz/Px)	320	315

Abbreviations: 2D GRE, 2-dimensional gradient echo; FA, flip angle; MRI, magnetic resonance imaging; T, tesla; ch., channel; TE, echo time; TR, repetition time.

**Table 2 sensors-21-06352-t002:** Transit times (Δt) and PWVs calculated from the sectional pulse waveforms.

Sections	Transit Time (ms)	PWV (m/s)
1	32.0	1.88
2	29.6	2.03
3	28.0	2.14
4	16.8	3.57
5	19.2	3.13
Mean ± SD	25.12 ± 6.71	2.55 ± 0.75

Note: The mean transit time and mean PWV were 25.23 ± 6.57 ms and 2.38 m/s for all the waveforms with 26 peaks, respectively. Abbreviations: PWV, pulse wave velocity; SD, standard deviation.

## Data Availability

The data presented in this study are available on request from the corresponding author.

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
