# Peer review of "Novel Technique to Measure Pulse Wave Velocity in Brain Vessels Using a Fast Simultaneous Multi-Slice Excitation Magnetic Resonance Sequence"

_sensors, 2021, doi:10.3390/s21196352_

Round 1

Reviewer 1 Report

The manuscript is well-organized and structured, and it includes all the necessary sections. 

I recommend the publication of this article, with the minor corrections mentioned above.

Reviewer 2 Report

The manuscript presents a novel method for measuring cerebral pulse wave velocity based on MRI measurements, with advantages over the currently available techniques. The results seem OK, but the methods are so poorly described (it described at all) that it is difficult to judge the soundness of the work. 

The entire section 2 needs to be improved, detailing the experimental setup and (especially) the signal processing methods. As it is, sections 2.1, 2.2  and 2.3 present the experiments performed, but the methods have not been described at all. What is a k-space? What is a keyhole? The authors need to explain in details how the data is acquired, which data is acquired, and how this data is processed in order to obtain the desired results.

Other minor issues are indicated in comments and highlights in the attached PDF.

Reviewer 3 Report

This study introduced a novel technique to measure the pulse wave velocity, which is called simultaneous multi-slice (SMS-K). It pointed out that SMS-K is simpler and has better accuracy than existing PWV techniques, where more data may be provided to support this claim.

  • Fix abbreviation problems in Abstract.
    1. In the 1st sentence in the Abstract, the SMS-K is not an abbreviation of “a novel PWV technique”. Please remove “(SMS-K)”.
    2. What the ‘K’ in “SMS-K” stands for?
    3. Please spell out PWV, pulse wave velocity, when it first showed up.
  • Please provide scale bar in all the MRI images in Figure 1, 2, 3.
  • It looks like that there was only one subject recruited in the SMS-K experiments. How representative is it? Do the authors have data from other subjects?
  • The authors pointed out that SMS-K has better accuracy than other techniques. However, they didn’t provide a quantitative metric for the accuracy. So why SMS-K is more accurate, given the limitations discussed in the last paragraph of Discussion?

Round 2

Reviewer 2 Report

The authors have properly addressed all my concerns, and have greatly improved the manuscript, with detailed descriptions of the methods. There are only some minor issues indicated in the attached PDF.

Author Response

Thank you very much for your favorable corrections. As the reviewer recommended, we corrected the typos and removed the underscores between number and unit. And the word FT in Figure 1 was changed to IFT as the reviewer commented.

Reviewer 3 Report

The authors fully addressed my comments. I recommend this manuscript for publication. Thanks!

Author Response

Thank you very much for your generous evaluation.
